# A Vancomycin HPLC Assay for Use in Gut Microbiome Research

Chenlin Hu,ᵃ Nicholas D. Beyda,ᵃ Kevin W. Gareyᵃ

ᵃUniversity of Houston College of Pharmacy, Houston, Texas, USA

**ABSTRACT**  The human microbiome project has revolutionized our understanding of the interaction between commensal microbes and human health. By far, the biggest perturbation of the microbiome involves use of broad-spectrum antibiotics excreted in the gut. Thus, pharmacodynamics of microbiome changes in relation to drug exposure pharmacokinetics is an emerging field. However, reproducibility studies are necessary to develop the field. A simple and fast high-performance liquid chromatography-photodiode array detector (HPLC) method was validated for quantitative fecal vancomycin analysis. Reproducibility of results were tested based on sample storage time, homogeneity of antibiotic within stool, and concentration consistency after lyophilization. The HPLC method enabled the complete elution of vancomycin within ~4.2 min on the reversed-phase C18 column under the isocratic elution mode, with excellent recovery (85% to 110%) over a 4-log, quantitative range (0.4–100 $\mu$g/mL). Relative standard derivations (RSD) of intra-day and inter-day results ranged from 0.4% to 5.4%. Using sample stool aliquots of various weights consistently demonstrated similar vancomycin concentrations (mean RSD: 6%; range: 2–16%). After correcting for water concentrations, vancomycin concentrations obtained after lyophilization were similar to the concentrations obtained from the original samples (RSD less than 10%). These methodologies establish sample condition standards for a quantitative HPLC to enable vancomycin pharmacokinetic studies with the human microbiome.

**IMPORTANCE** Research on antibiotic effect on the gut microbiome is an emerging field with standardization of research methods needed. In this study, a simple and fast high-performance liquid chromatography method was validated for quantitative fecal vancomycin analysis. Reproducibility of results were tested to standardize storage time, homogeneity of antibiotic within stool, and concentration consistency after lyophilization. These methodologies establish sample condition standards for a quantitative HPLC to enable vancomycin pharmacokinetic studies with the human microbiome.

**KEYWORDS**  vancomycin, glycopeptide, HPLC, clinical trial, metagenomics, drug distribution

The goal of the human microbiome project (HMP) is to characterize the human microbiome and analyze its role in health and disease (1). The HMP has advanced science on the systems biology of the gut microbiome, the most diverse and abundant microbiome in the human body. By far, the largest contributor to immediate changes in the human gut microbiome are broad spectrum antimicrobials that are excreted via the gastrointestinal tract. To be able to properly study the effect of antimicrobials on the gut microbiome, pharmacokinetic studies using validated assays in the correct biologic matrix will be required. The most common antibiotic assessed to date for microbiome disruption has been the glycopeptide antibiotic, vancomycin due to its common use in patients with *Clostridioides difficile* infection (CDI) (2). Chemically, vancomycin is a water-soluble, tricyclic glycosylated peptide (3), is absorbed minimally from the human gastrointestinal (GI) tract and mainly excreted into the stool (4, 5). Oral vancomycin PK studies have investigated the association of vancomycin dose with outcomes (6), and the compositional change in

Address correspondence to Kevin W. Garey, kgarey@uh.edu.

The authors declare no conflict of interest.

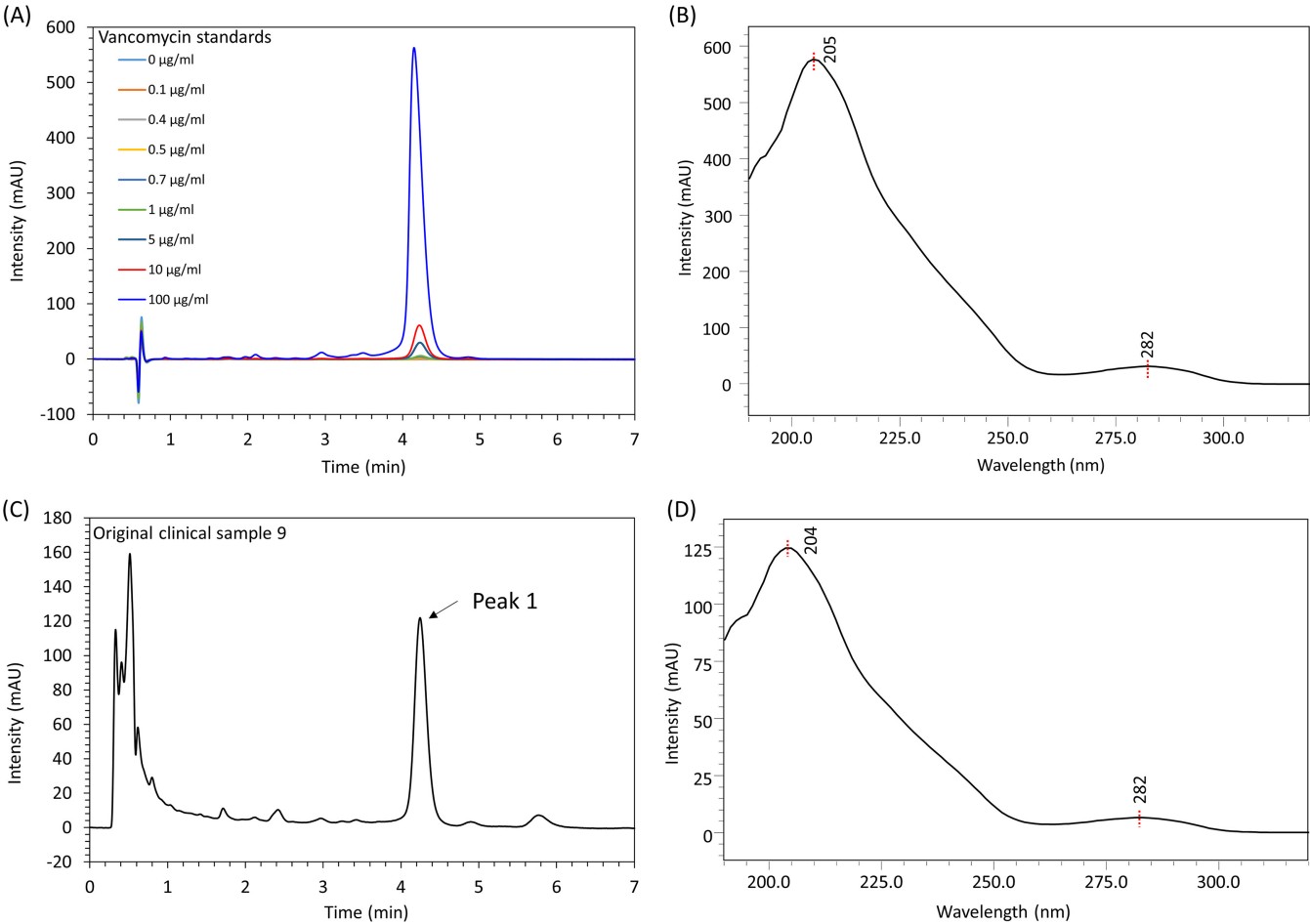

**FIG 1** Chromatographic profiles of vancomycin extracted from original sample. (A) Chromatographic profiles of the vancomycin standard solutions (0, 0.1, 0.4, 0.7, 1, 5. 10, 100 μg/mL, injection volume: 5 μL, retention time: ~ 4.2 min) monitored at 205 nm. (B) The UV absorption pattern of the vancomycin standard (100 μg/mL). (C) The representative chromatographic profile of the fecal extract (diluted by 10-fold) of an original vancomycin-containing clinical sample 9 monitored at 205 nm, Peak 1 with a retention time at 4.244 min was identified to be the vancomycin at a concentration of 20.1 μg/mL in the diluted fecal extract solution. (D) The UV absorption pattern of Peak 1 (identified as vancomycin) that was consistent with that of standard vancomycin in Fig. 1B.

human gut microbiome (7, 8). However, most vancomycin assays developed to date have been semiquantitative and not suitable for quantitative pharmacokinetic analyses (4, 5, 9, 10). A quantitative liquid chromatography-mass spectrometry has been used in clinical trials, but this assay was never validated (11). Thus, a validated vancomycin assay with appropriate reproducibility studies has not been published.

Our research group recently completed a phase I healthy volunteer study of ibezapolstat, a DNA polymerase IIIC inhibitor in clinical development for the treatment of CDI (7). Healthy subjects given vancomycin were recruited as a comparator and fecal samples were sent to the University of Houston Research lab for HPLC analysis. Leading up to and during the conduct of the study, it was noted that lack of a validated assay for vancomycin PK and gut microbiome analysis along with several needed reproducibility studies had not been undertaken. The purpose of this project was to validate our newly developed fecal vancomycin HPLC method and undertake several reproducibility experiments to help inform future microbiome studies. Specific questions for reproducibility experiments were stability of vancomycin samples in stool over time, whether vancomycin was equally distributed in the samples, and correlation between original and lyophilized samples.

## RESULTS

**HPLC method.** Under the present isocratic elution mode vancomycin was eluted from the reversed-phase C18 column at ~4.2 min (Fig. 1A); the UV absorption spectrum of

**TABLE 1** Inter-day vancomycin concentrations in clinical and spiked fecal samples

| Clinical samples | | | |
|---|---|---|---|
| Sample no. | Day 0, vancomycin concn[a] | Day 48, vancomycin concn | Inter-day RSD (%) |
| 5 | 269 ± 3 | 259 ± 14 | 2.60 |
| 14 | 1,211 ± 8 | 1,220 ± 11 | 0.90 |
| 9 | 1,897 ± 7 | 1,905 ± 12 | 0.60 |
| | | | |
| Spiked samples | | | |
| Spiked vancomycin concn, $\mu$g/g | Day 0, HPLC-PDA signal | Day 7, HPLC-PDA signal | Inter-day RSD (%) |
| 0.4 | 27,368 | 28,525 | 2.93 |
| 0.5 | 36,414 | 34,307 | 4.21 |
| 0.7 | 49,159 | 48,141 | 1.48 |
| 1 | 70,925 | 73,311 | 2.34 |
| 10 | 739,333 | 739,226 | 0.01 |
| 100 | 7,232,766 | 7,206,891 | 0.25 |

[a]Values represent Mean ± SD, $\mu$g/g.

vancomycin showed the presence of a major peak at around 205 nm and a minor peak at around 282 nm (Fig. 1B), consistent with the UV spectrum of vancomycin standard (12). Therefore, absorption at 205 nm was used for vancomycin quantification. Injecting 5 $\mu$L of the vancomycin standard in the matrix of acetonitrile-water mixture, the limit of detection (LOD) and limit of quantification (LOQ) was 0.1 and 0.4 $\mu$g/mL, respectively. The method exhibited a wide working quantification range (0.4–100 $\mu$g/mL) with the good linearity ($R^2 > 0.99$: linear fit equation: $Y = 72,271X – 7,070$ where X = vancomycin concentration and Y = HPLC peak area). Other chromatograms are shown in the Fig. S1-S9.

**Reproducibility studies. (i) Vancomycin stability in stool samples over time.** Vancomycin concentrations from spiked samples were relatively unchanged between the 7-day time period at concentration ranges from 0.4–100 $\mu$g/g of stool (RSD range: 0.01–4.2%). Identical aliquots ($n = 3$) of three clinical samples assayed 50 days apart also demonstrated good stability within the aliquots and very little difference between vancomycin concentrations between the two time periods. The intra-day relative standard derivations (RSDs) ranged from 0.6–2.6% for the three clinical samples (Table 1).

**(ii) Vancomycin distribution in stool samples.** The recovery rates of vancomycin standard (1, 10, 50, and 100 $\mu$g) added into the fecal samples of differing weights (32.9-83.2 mg) all averaged above 85% with a RSD less than 10% for all analyses (Table 2). Vancomycin concentrations among aliquots of identical clinical samples were also distributed evenly among aliquots with RSD all below 16% and most (9 of 10 tested samples) with an RSD of 10% or less (Table 3).

**(iii) Vancomycin concentrations in original and lyophilized samples.** Nine samples were divided into 2–3 aliquots each for HPLC analysis using the original sample or a lyophilized sample (Table 4). Concentrations between aliquots for both original and lyophilized samples were below 15% RSD for all samples. Water concentration in the original sample

**TABLE 2** Recovery rates of four levels of vancomycin (1, 10, 50, and 150 $\mu$g) added to reference stool sample

| Vancomycin spiked ($\mu$g) | Spiked aliquot (wet wt, mg) | Measured vancomycin, $\mu$g | Recovery (mean ± SD, %) | RSD[a] (%) |
|---|---|---|---|---|
| | A1 (32.9) | 1.2 | | |
| 1 | A2 (49.6) | 1.0 | 110 ± 10 | 9 |
| | A3 (45.9) | 1.1 | | |
| | A4 (78.2) | 9.1 | | |
| 10 | A5 (65.4) | 8.5 | 85 ± 5 | 6 |
| | A6 (53.9) | 8.0 | | |
| | A7 (83.2) | 43.7 | | |
| 50 | A8 (47.2) | 46.3 | 88 ± 5 | 6 |
| | A9 (68.9) | 41.2 | | |
| | A10 (58.1) | 139.2 | | |
| 150 | A11 (43.5) | 141.9 | 93 ± 1 | 1 |
| | A12 (62.4) | 139.5 | | |

[a]RSD, relative standard derivation.

**TABLE 3** Fecal vancomycin levels from original clinical samples

| Sample | Subject | Age | Treatment | Day | Vancomycin concn ($\mu$g/g) | | | Avg | RSD[c] (%) |
|---|---|---|---|---|---|---|---|---|---|
| | | | | | Aliquot 1 | Aliquot 2 | Aliquot 3 | | |
| 1 | A | 25 | Reference | | ND[b] (58.9) | ND (40.4) | ND (39.3) | | |
| 2 | B | 32 | Reference | | ND (23.3) | ND (31.6) | ND (42.9) | | |
| 3 | H | 36 | Vanco[a] | 2 | ND (37.2) | ND (41.9) | ND (52.6) | | |
| 4 | G | 29 | Vanco | 2 | ND (43.0) | ND (65.2) | ND (99.6) | | |
| 5 | H | 36 | Vanco | 5 | 269 (37.8) | 259 (52.9) | 259 (33.0) | 262 | 2 |
| 6 | E | 33 | Vanco | 8 | 1863 (80.8) | 2033 (40.7) | 1660 (106.5) | 1852 | 10 |
| 7 | F | 38 | Vanco | 8 | 1831 (60.1) | 1877 (55.7) | 2057 (48.1) | 1922 | 6 |
| 8 | E | 33 | Vanco | 9 | 1953 (68.8) | 1868 (91.7) | 1920 (61.5) | 1914 | 2 |
| 9 | F | 38 | Vanco | 9 | 1897 (106.0) | 1964 (68.4) | 1973 (80.2) | 1945 | 2 |
| 10 | D | 28 | Vanco | 10 | 1695 (63.4) | 1947 (34.7) | 1775 (57.0) | 1806 | 7 |
| 11 | D | 28 | Vanco | 11 | 1759 (68.7) | 1801 (62.7) | 1891 (64.0) | 1817 | 4 |
| 12 | F | 38 | Vanco | 11 | 1469 (78.2) | 1920 (47) | 1488 (54.2) | 1626 | 16 |
| 13 | C | 27 | Vanco | 12 | ND (74.9) | ND (61.6) | ND (90.5) | | |
| 14 | F | 38 | Vanco | 12 | 1211 (129.4) | 1162 (60.8) | 1070 (74.6) | 1148 | 6 |
| 15 | D | 28 | Vanco | 12 | 1187 (88.3) | 1111 (90.1) | 1200 (59.4) | 1166 | 4 |
| 16 | G | 29 | Vanco | 32 | ND (78.2) | ND (54.4) | ND (40.0) | | |

[a]Vanco, oral vancomycin dose 125 mg, 4 times daily for 10 days.
[b]ND, not detectable.
[c]RSD, relative standard derivation; the number shown in the brackets indicates the wet weight (mg) of each aliquot used.

averaged 78 $\pm$ 3% for lyophilized stool samples. After correcting for water concentrations, vancomycin concentrations obtained after lyophilization was similar to the concentrations obtained from the original samples (RSD less than 10%).

**(iv) Comparison with previously developed orthophosphoric acid-based extraction method.** The ratio of fecal vancomycin concentration between our method and the previously published method ranged from 91% to 98% ($n = 6$, mean = 93%, Table S1) demonstrating good performance between the two methods.

## DISCUSSION

Antibiotic disruption studies of the microbiome typically involve a pharmacokinetic analysis of the antibiotic along with metagenomics of functional microbiome studies (13). Given the relative newness of the field, there is still a considerable amount of work needed to standardize the workflow for antibiotic and microbiome studies. In a recently completed phase I clinical trial, we developed a rapid HPLC method for the quantitative determination of vancomycin in stool (7). During this study several unknowns on the reproducibility of microbiome and pharmacokinetic studies were identified. It was these prior experiences that helped inform the objectives of this study. The aim for this current study was to validate our HPLC method for fecal vancomycin quantitation, including stability of vancomycin over time, to assess whether vancomycin was equally distributed in the samples, and comparison between original and lyophilized stool sample vancomycin concentrations.

Our previous clinical trial results and these validation studies confirm that this method is sensitive, accurate, stable, and suitable for fecal vancomycin analysis. The LOD value of this HPLC method is comparable to those (0.1-1 $\mu$g/mL) of the recently developed HPLC methods for systemic vancomycin concentrations (14, 15). Given the typically high fecal vancomycin level (>100 $\mu$g/mL) generated after standard oral vancomycin administration (5, 16), our present method is sufficiently sensitive for the clinical fecal vancomycin analysis. Moreover, the quantitative range of our new method (0.4–100 $\mu$g/mL) can be diluted allowing for analysis at much higher concentrations. Additionally, our extraction method for total fecal vancomycin was comparable to a previously described orthophosphoric acid-based extraction method (17). Finally, vancomycin concentrations are stable over time up to 49 days studied in these experiments. Clinically, we were able to demonstrate that vancomycin persists for several days following the end of the 10-day dosing regimen and also that concentrations remain low in the stool several days after starting therapy. These observations help justify the need for research related to the pharmacokinetics of vancomycin in the gut.

**TABLE 4** Comparison of vancomycin concentrations from original versus lyophilized clinical samples

| Subject | Sample | Aliquot | Original samples | | | | Lyophilized samples | | | | | | | Comparison RSD, original versus lyophilized (%) |
|---|---|---|---|---|---|---|---|---|---|---|---|---|---|---|
| | | | Wt[a] | Vanco conc[b] | Mean[c] | RSD,[d] % | Wt | Vanco conc | Mean[e] | RSD, % | Water percentage (%) | Water corrected concn | Mean | |
| D | 10 | 1 | 63.4 | 1,695 | 1,806 | 7.1 | 41.2 | 7,501 | 7,206 | 5.8 | 76 | 1,800 | 1,729 | 3.1 |
| D | 10 | 2 | 34.7 | 1,947 | | | 75.3 | 6,911 | | | 76 | 1,659 | | |
| D | 10 | 3 | 57 | 1,775 | | | NA[e] | | | | | | | |
| D | 11 | 1 | 68.7 | 1,759 | 1,817 | 3.7 | 51.9 | 7,688 | 7,437 | 11.9 | 74 | 1,999 | 1,934 | 4.2 |
| D | 11 | 2 | 62.7 | 1,801 | | | 36.8 | 8,166 | | | 74 | 2,123 | | |
| D | 11 | 3 | 64 | 1,891 | | | 30 | 6,457 | | | 74 | 1,679 | | |
| D | 15 | 1 | 88.3 | 1,187 | 1,166 | 4.1 | 41.6 | 5,058 | 4,973 | 1.5 | 75 | 1,265 | 1,243 | 4.6 |
| D | 15 | 2 | 90.1 | 1,111 | | | 38.5 | 4,941 | | | 75 | 1,235 | | |
| D | 15 | 3 | 59.4 | 1,200 | | | 44.1 | 4,921 | | | 75 | 1,230 | | |
| E | 6 | 1 | 80.8 | 1,863 | 1,852 | 10.1 | 31 | 9,206 | 9,658 | 6.6 | 79 | 1,933 | 2,028 | 6.3 |
| E | 6 | 2 | 40.7 | 2,033 | | | 58.2 | 10,109 | | | 79 | 2,123 | | |
| E | 6 | 3 | 106.5 | 1,660 | | | NA | | | | | | | |
| E | 8 | 1 | 68.8 | 1,953 | 1,914 | 2.2 | 28.7 | 10,116 | 9,781 | 4.9 | 78 | 2,226 | 2,152 | 8.4 |
| E | 8 | 2 | 91.7 | 1,868 | | | 39.4 | 9,445 | | | 78 | 2,078 | | |
| E | 8 | 3 | 61.5 | 1,920 | | | NA | | | | | | | |
| F | 7 | 1 | 60.1 | 1,831 | 1,922 | 6.2 | 44.6 | 8,614 | 8,991 | 5.9 | 75 | 2,154 | 2,248 | 10.8 |
| F | 7 | 2 | 55.7 | 1,877 | | | 28.2 | 9,367 | | | 75 | 2,342 | | |
| F | 7 | 3 | 48.1 | 2,057 | | | NA | | | | | | | |
| F | 9 | 1 | 106 | 1,897 | 1,945 | 2.1 | 26.7 | 9,745 | 9,220 | 4.9 | 77 | 2,241 | 2,121 | 6.2 |
| F | 9 | 2 | 68.4 | 1,964 | | | 43.3 | 8,981 | | | 77 | 2,066 | | |
| F | 9 | 3 | 80.2 | 1,973 | | | 60.8 | 8,935 | | | 77 | 2,055 | | |
| F | 12 | 1 | 78.2 | 1,469 | 1,626 | 15.7 | 12.7 | 9,026 | 8,184 | 13.2 | 82 | 1,625 | 1,473 | 6.8 |
| F | 12 | 2 | 47 | 1,920 | | | 32.6 | 8,563 | | | 82 | 1,541 | | |
| F | 12 | 3 | 54.2 | 1,488 | | | 49.2 | 6,962 | | | 82 | 1,253 | | |
| F | 14 | 1 | 129.4 | 1,211 | 1,148 | 6.2 | 19 | 6,748 | 6,120 | 8.9 | 81 | 1,282 | 1,163 | 1.0 |
| F | 14 | 2 | 60.8 | 1,162 | | | 37.1 | 5,810 | | | 81 | 1,104 | | |
| F | 14 | 3 | 74.6 | 1,070 | | | 56.2 | 5,803 | | | 81 | 1,103 | | |

[a]Wt, weight (mg) of each aliquot used.
[b]Vanco conc, vancomycin concentration ($\mu$g/g).
[c]Mean, the average value of vancomycin concentrations ($\mu$g/g) in the different aliquots of the same sample.
[d]RSD, relative standard derivation.
[e]NA, not analyzed.

During the phase I clinical trial mentioned above, the pharmacokinetics of ibezapolstat was evaluated using a commercial laboratory. They required the entire stool in order to homogenize the sample for accurate pharmacokinetics results as the heterogeneity of antibiotics in stool samples has not been studied. This differs from blood, urine, or other matrices in which an almost constant supply is present. However, as bowl movements only occur once per day or less, use of an entire sample for pharmacokinetic analyses excludes other microbiome evaluations from that particular day or time-period. In this study, we demonstrated that vancomycin was uniformly distributed in the sample allowing for aliquots of stool to be sent to pharmacokinetic analyses allowing other aliquots to be used for other purposes. Likewise, sample preparation for functional microbiome analyses (for example, bile acid analysis) often involves a lyophilization step. After correcting for water percentage, we also demonstrated similar results using lyophilized samples compared to the original sample concentration. Samples were also distributed uniformly in lyophilized samples as well providing evidence that aliquots of lyophilized samples can also be used. There was a small amount of heterogeneity between samples for both original and lyophilized. For this reason, we will use 2 aliquots for pharmacokinetics evaluations with our vancomycin HPLC in the future. Vancomycin is a hydrophilic drug. We will need to perform similar analyses to demonstrate homogeneity of drug distribution for other, nonabsorbable hydrophobic drugs as well as systemically absorbed antibiotics that are excreted in the colon.

In conclusion, this study validated our quantitative HPLC vancomycin stool assay and also established sample condition standards to further vancomycin pharmacokinetic studies with the human microbiome.

## MATERIALS AND METHODS

**Materials.** Acetonitrile and water (LC-MS-grade) as well as trifluoreacetic acid (TFA) were purchased from VWR International and ThermoFisher Scientific (USA), respectively. Vancomycin hydrochloride powder (United States Pharmacopeia Reference Standard) was purchased from Millipore-Sigma (USA) and was dissolved in the mixture of acetonitrile and water (10:90 [v:v], 0.1% TFA) to prepare the stock solution (10 mg/mL) and the working standard solutions of vancomycin (0.1, 0.4, 0.5, 0.7, 1, 5, 10, 100 $\mu$g/mL).

**Clinical fecal sample collection and pretreatment.** A total of 16 human fecal samples from ongoing IRB-approved healthy volunteer studies with vancomycin were used (7). The samples involved a total of 9 healthy male, nonsmoker, omnivorous volunteers (ages:25–38 years old) given placebo used for blank reference samples or oral vancomycin 125 mg 4 times per day for 10 days with stool samples obtained at baseline, during therapy, and at a follow-up visit performed at day 32.

**Extraction of vancomycin from stool samples.** An aliquot of stool and 1 mL of extraction solvent (acetonitrile and water, 10:90 [vol/vol], 0.1% TFA) was added into 2-mL centrifuge tube and shaken (Vortex-Genie 2 Shaker, Scientific Industries, USA) for 10 s, ultrasonicated for 5 min (Multool ultrasonic cleaner, Model: TH-030A), and centrifuged at 10,000 g for 3 min (Centrifuge 5804 R, Eppendorf). The supernatant was collected and diluted by 10-fold using the extraction solvent and was stored at −80°C prior to HPLC analysis.

**High performance liquid chromatography (HPLC) analysis.** Five $\mu$L of vancomycin standard working solution or fecal vancomycin extract was injected into the Shimadzu Nexera-i LC-2040C 3D Plus High Performance Liquid Chromatography (HPLC) system. Vancomycin was separated on the analytical Shimadzu Nexcol analytical column (C18, 1.8 $\mu$m, 50 × 2.1 mm) connected with a Shimadzu Nexcol guard column (C18, 5 $\mu$m, 3 mm ID x 5 mm L). The mobile phase consisted of a mixture of acetonitrile and water, (10:90 [v:v], 0.1% TFA). The flow rate and the temperature of column oven were set at 0.3 mL/min and 40°C, respectively. Fecal vancomycin was quantified based on the standard calibration curve with vancomycin standard concentrations plotted against the corresponding HPLC peak areas. The limit of detection (LOD) and limit of quantification (LOQ) were measured at a vancomycin level that yielded the signal to noise ratio (S/N) of 3 and 10, respectively (18).

**Reproducibility experiments.** Stability of vancomycin samples in stool over time was assessed by spiking known vancomycin concentrations into reference (vancomycin-free) stool samples and assayed twice separated by a 7-day interval. Second, vancomycin concentrations from three aliquots of three clinical samples were split in two and assayed 49 days apart. To assess whether vancomycin was equally distributed in stool samples, four different concentrations of vancomycin standard (1, 10, 50, and 100 $\mu$g) were spiked into a reference stool sample. Each sample was split into three different aliquots of differing weights (32.9–78.2 mg). The recovery rate of vancomycin calculated by dividing the recovered vancomycin amount by the vancomycin amount added (%). Second, clinical samples were split into three different aliquots of differing weights (23.3–129.4 mg) and assayed. Differences in concentrations were compared by RSD. To compare concentrations of vancomycin between original and lyophilized fecal samples, vancomycin concentrations from original samples and the corresponding lyophilized sample was assessed correcting for the percent water in the original sample.

**Comparison with the previously published orthophosphoric acid-based extraction method.** The performance of our extraction method was compared with an orthophosphoric acid-based extraction method, which was previously validated for vancomycin analysis (17). Briefly, six samples were chosen with two aliquots were prepared for each sample. The aliquots were mixed with 971 $\mu$L of water and 29 $\mu$L of 85% orthophosphoric acid according to the composition of orthophosphoric acid-based extraction solvent developed by Berthoin et al. (2009) (17). After vortexing, ultrasonification, and centrifuging, 50 $\mu$L of supernatant was diluted 20-fold and then applied to the HPLC system for total fecal vancomycin analysis. The ratio of fecal vancomycin concentrations between using our extraction method and the previously published orthophosphoric acid-based method was compared.

## SUPPLEMENTAL MATERIAL

Supplemental material is available online only.
**SUPPLEMENTAL FILE 1**, PDF file, 0.4 MB.

## ACKNOWLEDGMENTS

This work was supported by the National Institutes of Health NIAID (U01AI124290). We acknowledge Caroline Loveall for technical assistance during the study.

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
