## [Reviewer comments · Microbiology Spectrum]

Microbiology Spectrum

A Vancomycin HPLC Assay for Use in Gut Microbiome Research

Chenlin Hu, Nicholas Beyda, and Kevin Garey

Corresponding Author(s): Kevin Garey, University of Houston

Review Timeline:

Submission Date:	September 27, 2021
Editorial Decision:	December 1, 2021
Revision Received:	January 5, 2022
Editorial Decision:	March 6, 2022
Revision Received:	March 29, 2022
Accepted:	April 17, 2022

Editor: Adriana Rosato

Reviewer(s): The reviewers have opted to remain anonymous.

Transaction Report:

DOI: <https://doi.org/10.1128/spectrum.01688-21>

November 30, 2021

Dr. Kevin W Garey
University of Houston
Pharmacy Practice & Translational Research
4849 Calhoun Road
Houston, TX 77084

Re: Spectrum01688-21 (A Vancomycin HPLC-PDA Assay for Use in Gut Microbiome Research)

Dear Dr. Kevin W Garey:

Thank you for submitting your manuscript to Microbiology Spectrum.

Your manuscript has been reviewed by two experts in the field. Based on their criticism as well as my evaluation the manuscript requires a careful revision, mainly critical points that relate to the HPLC methodology and how vancomycin was extracted from the stool sample.

I'm also recommending reconsidering to restructure the study as a short note.

My recommendations to the authors are:

1-Clearly state whether Fig 1 is the result of the original or lyophilized sample.

2-How the authors explain that Vancomycin levels were not detected on subject G at both time points; Table 3 at days 2 and 32

3- Considering that this study offers a newly developed method for fecal vancomycin quantification, I will recommend for validation purposes to provide discussions/ examples of additional fecal extract chromatograms (e.g supplemental data)

Link Not Available

Sincerely,

Adriana Rosato

Journals Department
Reviewer comments:

Reviewer #1 (Comments for the Author):

Comment 1:

Page 1, line 1: In the title, a vancomycin HPLC-PDA assay was presented. In my opinion, only a single wavelength 205nm was

chosen for measuring vancomycin levels. I would remove "PDA" from the title.

Comment 2:

Page 2, line 25: The authors used "isostatic" elution mode. The "isocratic" elution mode is the correct term in HPLC.

Comment 3:

In the section of materials and methods, the extraction of vancomycin from fecal samples was not provided and detailed.

Comment 4:

Page 6, line 129: The authors stated "Water concentration averaged 78 percent for lyophilized stool samples. This sentence did not make sense, for we know that after the lyophilization process, the final residual water content in the stool is less than 5%.

Comment 5:

The authors should use an internal standard method by adding an internal standard to the stool samples prior to extracting vancomycin from the samples. In this way, human errors during the correction of water concentrations would be largely reduced.

Reviewer #2 (Comments for the Author):

The manuscript represents premature work. The Material and Methods section does not adequately provide information on how vancomycin was extracted from the stool sample. This is critical because the significant mass percentage of stool is represented by bacterial cells. Hence, orally administered vancomycin bound to the cell wall of bacteria present significant portions that are not detected by the method. Furthermore, the administered vancomycin affects the number of bacteria present in the gut and thereby altering the unbound free vancomycin concentration.

Staff Comments:

Preparing Revision Guidelines

Please return the manuscript within 60 days; if you cannot complete the modification within this time period, please contact me. If you do not wish to modify the manuscript and prefer to submit it to another journal, please notify me of your decision immediately so that the manuscript may be formally withdrawn from consideration by Microbiology Spectrum.

Editor Comments

1-Clearly state whether Fig 1 is the result of the original or lyophilized sample.

**Have modified the figure title accordingly

2-How the authors explain that Vancomycin levels were not detected on subject G at both time points; Table 3 at days 2 and 32

**This is an interesting observation, especially day 2. There is only a 10 day dosing interval so the day 32 sample not having detectable vancomycin is understandable. Day 2 is a little less intuitive but is likely dependent on transit time of the vancomycin capsule to be excreted in the feces. Have mentioned both of these observations in the discussion.

3- Considering that this study offers a newly developed method for fecal vancomycin quantification, I will recommend for validation purposes to provide discussions/ examples of additional fecal extract chromatograms (e.g supplemental data)

**Have added additional chromatograms as supplemental data as requested.

Reviewer comments:

Reviewer #1 (Comments for the Author):

Comment 1:

Page 1, line 1: In the title, a vancomycin HPLC-PDA assay was presented. In my opinion, only a single wavelength 205nm was chosen for measuring vancomycin levels. I would remove "PDA" from the title.

**Removed PDA as requested.

Comment 2:

Page 2, line 25: The authors used "isostatic" elution mode. The "isocratic" elution mode is the correct term in HPLC.

**Yes, good spot. Thank you.

Comment 3:

In the section of materials and methods, the extraction of vancomycin from fecal samples was not provided and detailed.

**Chenlin

Comment 4:

Page 6, line 129: The authors stated "Water concentration averaged 78 percent for lyophilized stool samples. This sentence did not make sense, for we know that after the lyophilization process, the final residual water content in the stool is less than 5%.

**Have changed this sentence to improve clarity

Comment 5:

The authors should use an internal standard method by adding an internal standard to the stool samples prior to extracting vancomycin from the samples. In this way, human errors during the correction of water concentrations would be largely reduced.

**This is an interesting thought. In this study, we either used known, spiked vancomycin stool samples or clinical samples from healthy subjects who only took oral vancomycin as part of an ongoing phase I study. For these reasons, we did not feel the need to add an internal standard. In addition, our correction for water concentrations allowed us to be quite close to the spiked concentrations (or our spiking experiments). We did contemplate on adding an internal standard for future studies in patients in which other small molecules may interfere with the HPLC readings.

Reviewer #2 (Comments for the Author):

The manuscript represents premature work. The Material and Methods section does not adequately provide information on how vancomycin was extracted from the stool sample. This is critical because the significant mass percentage of stool is represented by bacterial cells. Hence, orally administered vancomycin bound to the cell wall of bacteria present significant portions that are not detected by the method. Furthermore, the administered vancomycin affects the number of bacteria present in the gut and thereby altering the unbound free vancomycin concentration.

**Thank you for these comments. We have added a section on how vancomycin was extracted from the stool sample based on this comment and comments from reviewer #2.

March 6, 2022

Dr. Kevin W Garey
University of Houston
Pharmacy Practice & Translational Research
4849 Calhoun Road
Houston, TX 77084

Re: Spectrum01688-21R1 (A Vancomycin HPLC Assay for Use in Gut Microbiome Research)

Dear Dr. Kevin W Garey:

Your manuscript has been reviewed and there is still an important point to address notably in what concerns the methodology as highlighted by Rev1.

In addition, I would like to apologize for the time that this revision has taken, I have been waiting for Rev 2 critics. The decision is based on Rev 1 and my revision. I would be happy to review your final version.

Link Not Available

Sincerely,

Adriana Rosato

Journals Department
Reviewer comments:

Reviewer #1 (Comments for the Author):

Comment 1:

Although vancomycin is a hydrophilic drug, the authors should compare the extraction solvent used in the current method with a previous extraction solvent published in the International Journal of Antimicrobial Agents 34 (2009) 555-560. The published report showed that there was a wide variation in free to total vancomycin serum concentrations in patients (range 12-100%)

treated for gram-positive infections. This is critical because vancomycin bound to the wall of bacteria present significant portions that could not be detected by the current method.

Comment 2:

Ultracentrifugation devices were not provided in the Material and Methods section.

Comment 3:

(Table 3) is missing on page 3 line 144.

Comment 4:

What is "nad" on page 8 line 158?

Staff Comments:

Preparing Revision Guidelines

Please return the manuscript within 60 days; if you cannot complete the modification within this time period, please contact me. If you do not wish to modify the manuscript and prefer to submit it to another journal, please notify me of your decision immediately so that the manuscript may be formally withdrawn from consideration by Microbiology Spectrum.

Editor Comments

Comment 1:

Although vancomycin is a hydrophilic drug, the authors should compare the extraction solvent used in the current method with a previous extraction solvent published in the International Journal of Antimicrobial Agents 34 (2009) 555-560. The published report showed that there was a wide variation in free to total vancomycin serum concentrations in patients (range 12-100%) treated for gram-positive infections. This is critical because vancomycin bound to the wall of bacteria present significant portions that could not be detected by the current method.

**Have added an extra experiment comparing the two techniques. Thanks for this suggestion.

Comment 2:

Ultracentrifugation devices were not provided in the Material and Methods section.

**Have added this into the Methods section.

Comment 3:

(Table 3) is missing on page 3 line 144.

**Thank you, good spot

Comment 4:

What is "nad" on page 8 line 158?

**That was a typo, now corrected (and)

April 17, 2022

Dr. Kevin W Garey
University of Houston
Pharmacy Practice & Translational Research
4849 Calhoun Road
Houston, TX 77084

Re: Spectrum01688-21R2 (A Vancomycin HPLC Assay for Use in Gut Microbiome Research)

Dear Dr. Kevin W Garey:

I'm please to inform that your manuscript has been accepted, and I am forwarding it to the ASM Journals Department for publication. You will be notified when your proofs are ready to be viewed.

Sincerely,

Adriana Rosato
Editor, Microbiology Spectrum
